# Peering into the Brain’s Estrogen Receptors: PET Tracers for Visualization of Nuclear and Extranuclear Estrogen Receptors in Brain Disorders

**DOI:** 10.3390/biom13091405

**Published:** 2023-09-18

**Authors:** Shokouh Arjmand, Dirk Bender, Steen Jakobsen, Gregers Wegener, Anne M. Landau

**Affiliations:** 1Translational Neuropsychiatry Unit, Department of Clinical Medicine, Aarhus University, 8200 Aarhus, Denmark; wegener@clin.au.dk; 2Department of Nuclear Medicine and PET Center, Aarhus University Hospital, 8200 Aarhus, Denmark; dirkbend@rm.dk (D.B.); steejako@rm.dk (S.J.)

**Keywords:** PET, estrogen receptors, radioligand, ERα, ERβ, GPER, positron emission tomography, genomic estrogen signaling, non-genomic estrogen signaling, membrane estrogen receptors

## Abstract

Estrogen receptors (ERs) play a multitude of roles in brain function and are implicated in various brain disorders. The use of positron emission tomography (PET) tracers for the visualization of ERs’ intricate landscape has shown promise in oncology but remains limited in the context of brain disorders. Despite recent progress in the identification and development of more selective ligands for various ERs subtypes, further optimization is necessary to enable the reliable and efficient imaging of these receptors. In this perspective, we briefly touch upon the significance of estrogen signaling in the brain and raise the setbacks associated with the development of PET tracers for identification of specific ERs subtypes in the brain. We then propose avenues for developing efficient PET tracers to non-invasively study the dynamics of ERs in the brain, as well as neuropsychiatric diseases associated with their malfunction in a longitudinal manner. This perspective puts several potential candidates on the table and highlights the unmet needs and areas requiring further research to unlock the full potential of PET tracers for ERs imaging, ultimately aiding in deepening our understanding of ERs and forging new avenues for potential therapeutic strategies.

## 1. Introduction

Low success rates in drug development for psychiatric and neurological diseases can in part be attributed to an inadequate understanding of the intricate mechanisms and underlying molecular correlates that go awry in these diseases in vivo, and the dynamic complex nature of mental components such as cognition and emotion. Positron emission tomography (PET) imaging is among the effective investigational tools used to study the function of the brain and the pathophysiology associated with brain disorders in living subjects. Its ability to quantify molecular targets and measure the target occupancy of a drug molecule, as well as its biodistribution and pharmacokinetic parameters, phenomena that cannot be directly measured otherwise, has made PET a boon to studying the central nervous system (CNS). PET provides a selective and sensitive visualization of a target–tracer interaction in vivo, which could be associated with a given stage of a brain disorder, allow patient stratification, and permit the longitudinal investigation of responses to potential therapies.

Altered neurosteroid signaling has been associated with psychiatric and neurological disorders. Additionally, sex differences among different psychiatric and neurological disorders, such as autism spectrum disorders, anxiety disorders, major depressive disorders (MDD), Parkinson’s disease (PD), and Alzheimer disease (AD), have been consistently reported [1]. These observations may drive the identification of potential novel targets and etiological mechanistic approaches for treating these debilitating disorders.

Estrogens and their cognate estrogen receptors (ERs) have been shown to play diverse and pivotal roles in the CNS [2], yet our picture of their exact functions and the mechanisms through which they exert such actions in the brain is still fairly incomplete. Non-invasive techniques, such as PET imaging, can dramatically improve our understanding of the function of estrogen receptors in the brain and throughout different stages of a brain disease.

To develop efficient PET tracers for CNS applications, several criteria need to be met, with specific attention to their ability to cross the blood brain barrier (BBB). Moreover, what makes development of ERs PET tracers even more challenging is the presence of ERs both in the nucleus and at the plasma membrane, which are functionally distinct.

In this perspective, we first briefly review studies on estrogen and ERs in the brain and then highlight the importance of studying diverse subtypes of ERs in the brain, emphasizing the unique mode of action of each. We argue that there is a lack of selective CNS PET tracers for the detection of various ERs in the brain addressing the potential setbacks. To this end, we suggest several potential PET radioligand candidates.

## 2. Estrogen and Estrogen Receptors (ERs) in the Brain

Estrogens are steroid hormones with the phenolic ring A, the cyclohexane rings B and C, a methyl group in C13, and the ring D as the backbone. Depending on the number of hydroxyl groups in the chemical structure of the molecule, estrogens are categorized and named differently, showing different physico-chemical and pharmacokinetic properties. In terms of serum levels, during reproductive years, the most predominant estrogen is estradiol (E2), which is also the most potent estrogen. Estetrol (E4) is only produced during pregnancy by the human fetal liver, while estrone (E1), being the weakest estrogen in terms of potency, becomes the main postmenopausal estrogen [2]. Hence, the term “estrogen” most broadly refers to estradiol (E2), and in this context, we have also interchangeably used these terms.

Estrogen is locally synthesized de novo in the brain by neurons and astrocytes from cholesterol, as one of the neurosteroids [3]. The local metabolism of steroids can also result in the production of estrogens in the brain. Both peripherally and locally produced estrogens are able to target various regions of the brain and modulate sanguine cerebral functions and homeostasis through estrogen receptors found in different brain areas.

In the brain, aromatase is responsible for the conversion of androgen precursors into estrogens [4]. The hypothalamus has been demonstrated to have the highest levels of aromatase in the adult brain, especially in the preoptic area (POA) and ventromedial nucleus (VMN). Some other brain regions, including the amygdala, hippocampus, midbrain, cortical regions, cerebellum, and white matter also contain considerable amount of aromatase [5,6]. In these areas, the expression of aromatase is sex- and steroid-independent, whereas in the hypothalamus, higher concentrations of aromatase were found in males than females, as the amount of the circulating testosterone regulates aromatase expression and activity, which subsequently drives its conversion into E2 [5,6]. Such observations were initially investigated with molecular, immunohistochemistry, and electron microscopy techniques, and were further confirmed by PET imaging studies using radiolabeled aromatase inhibitors such as [^11^C]-vorozole and [^11^C]-cetrozole [7,8]. Another PET study using [^11^C]-vorozole revealed that menstrual cycle did not change the regional levels of aromatase [9].

Estrogen is thought to affect neural synaptic plasticity and spine synapse formation, migration, differentiation, and survival of newborn neurons, as well as proliferation of neural stem cells, and integration of the blood brain barrier [3,10,11,12,13,14,15,16]. It is also believed that both neuron- and astrocyte-derived estrogens play an essential role in neuroprotection and cognition [5,17]. These modulatory effects of estrogens are exerted through their interaction with different estrogen receptors in the brain, although some ligand-independent activation of estrogen receptors has been reported [18].

Autoradiography studies of mainly guinea pig, chick embryo, and rat brain tissues using [^3^H]-17β-estradiol [19,20,21,22,23], followed by in situ hybridization and immunohistochemistry techniques performed on rodent, macaque, guinea pig, and human brain tissues [24,25,26,27,28,29,30], have shed light on the presence of different types of estrogen receptors in various brain regions [31]. Based on preclinical studies, estrogen receptor alpha (ERα) was found to be abundantly expressed in the POA, bed nucleus stria terminalis (BNST), amygdala, periventricular nucleus (PV), VMN, as well as arcuate nucleus. Estrogen receptor beta (ERβ) has almost the same expression pattern, with higher levels in the POA, BNST, PV, and the supraoptic nuclei [28,31,32,33,34,35]. Both ERα and ERβ are also expressed in the cortex, hippocampus, midbrain, striatum, basal nucleus of Meynert, and diagonal band of Broca [34,36].

Co-localization studies demonstrated that corticotropin releasing hormone and insulin-like growth factor I expressing neurons and/or glia co-express both ERα and ERβ [24,37,38,39]. However, ERα is distinctly found to be co-localized in dopamine, norepinephrine, GABA, neuropeptide Y, proopiomelanocortin, somatostatin, galanin, and neurotensin containing neurons, while ERβ is primarily co-localized with gonadotropin releasing hormone, vasopressin, oxytocin, and midbrain serotonin expressing neurons [32,38,40,41,42,43,44,45,46,47,48,49,50,51,52,53,54,55].

Co-localization of the sex steroid receptors with the neurons that are involved in the pathophysiology of neuropsychiatric disorders could suggest their regulatory role in such neurons. It is also interesting to mention that areas of the brain frequently reported to be involved in neurodegenerative and psychiatric disorders often overlap.

## 3. Estrogen Receptors outside the Nucleus: Genomic vs. Non-Genomic Action of ERs

For the sake of clarity, going forward, we refer to the entire population of estrogen receptors with ERs (as a broad term that refers to estrogen receptors without any specification of the type (only as an abbreviation for estrogen receptors), nERs as a broad reference to nuclear estrogen receptors, which encompass nuclear ERα (nERα) and nuclear ERβ (nERβ), and finally mERs as a broad terminology to refer to all membrane estrogen receptors, including membrane ERα (mERα), membrane ERβ (mERβ), and G protein-coupled estrogen receptor 1 (GPER).

In 1975, the first evidence suggested the presence of estrogen receptors (ERα and ERβ), at the cell membrane of endometrial cells [56,57]. Up until then, ERs had been classified solely as nuclear receptors. These membrane-bound ERs, unlike the ERs that require translocation to the nucleus upon their activation, elicit an immediate response. Soon after, it was reported that both ERα and ERβ were also found at the cell membrane in the brain [58,59,60]. Several observations corroborated the rapid and distinct mode of action of these membrane-bound estrogen receptors (mERα and mERβ), as applications of E2 after the inhibition of transcription or use of membrane-impermeable E2 still exerted an effect leading to hyperpolarization of neural cells [61,62,63].

mERα and mERβ can form either homo- or hetero-dimers (predominantly homo-dimers), and interestingly truncated variants of the full-length ERs have also often been found to exist extranuclearly [64,65] (Figure 1) However, it is of note to mention that research on membrane-associated ERs has mainly focused on Erα.

In addition to mERα and mERβ, another ER belonging to the G protein-coupled family known as G protein-coupled estrogen receptor 1 (GPER), or GPR30, was later discovered. GPER is localized to the endoplasmic reticulum in neurons [66] and is found at the cell membrane [67] in some brain regions, predominantly in the hippocampus, hypothalamus, prefrontal cortex, and somatosensory cortex [68], as well as in the hypothalamus [69] of rats (Figure 1).

The classical signaling pathway that is activated through nuclear ERs is dependent on the formation of ER homo- or hetero-dimers, and its translocation to the nucleus and the subsequent binding to a specific part of the DNA to regulate the transcription of certain genes. This phenomenon, which is known as nuclear-initiated signaling or genomic signaling, is relatively slow due to its nature and requires the recruitment of some nuclear co-activators and regulatory proteins [31] (Figure 1).

ERs can also initiate rapid signal transduction through the activation of membrane-initiated estrogen signaling in the brain. mERα and mERβ are trafficked to the plasma membrane through palmitoylation (highly conserved cysteine palmitoylation sites were identified for both ERα and ERβ) and association with caveoline to be transported to the membrane caveolae, the functional signalosome [64,70,71,72]. The activation of mERs in the membrane raft gives rise to activation of various proximal kinases and production of secondary messengers leading to expansion of signal transduction. Unlike nuclear-initiated pathways, the non-genomic pathway is rapid, resulting in the initiation of a signaling cascade seconds to minutes after its activation affecting behavior and higher brain functions by induction of cellular changes (Figure 1).

## 4. Estrogen, ERs, and Link to Brain Disorders

Apart from playing a key role in the regulation of reproduction and socio-sexual behavior, as well as sexual differentiation [5], there are other diverse implications for estrogen in the brain (both peripheral or neuron/astrocyte-derived), including, but not limited to, neuroprotection, the modulation of synaptic plasticity and cognitive function, induction of growth hormone, facilitation of DNA repair, antioxidant activity, regulation of microglia, and cerebral blood flow [5,73]. For a detailed review of the literature, please see [5]. Brain-derived estrogen positively influences both neuronal and astrocytic functions, reducing neuronal damage and preserving the brain’s cognitive ability [5].

E2 is also involved in synaptic transmission, potentiating glutamatergic, serotonergic, and dopaminergic transmission, and suppressing GABAergic neurotransmission [74]. E2 increases serotonin and dopamine availability via the induction of their synthesis, the inhibition of their degradation and reuptake, and upregulation of their corresponding receptors [75,76,77,78,79,80].

The agonism of ERα and ERβ can bring about a rapid increase in the influx of Ca^2+^ in neurons with the consequence of MAPK and ERK phosphorylation, which promotes neuroprotection [81]. ERα plays a protective role against neurotoxicity associated with hyper-activation of the glutamatergic system [81,82]. Gene knockdown animal studies of ERβ have pointed out the role of this receptor in the survival and differentiation of neurons. Abnormal neural morphology during brain development, reduced quantity of cortical neurons and their migration, and enhanced apoptosis were observed in ERβ knockdown rodents [83,84]. The activation of ERβ is also associated with elevated protein levels of the brain-derived neurotrophic factor (BDNF) important for neuronal function and plasticity [85,86].

Furthermore, GPER agonists were reported to enhance dendritic spine density in the hippocampus taking part in the estrogen-related mediation of learning and memory through rapid signaling. The BDNF expression level also increases after the activation of GPER, promoting synaptic plasticity. The PI3K/Akt/MAPK pathway is regulated via GPER exerting neuroprotective action [87].

Overall, estrogens, through their interactions with GPER, ERα, and ERβ, play critical roles in neuroprotection, synaptic transmission, neuronal survival, and synaptic plasticity, contributing to the modulation of brain function and potential applications in neurodegenerative, neurodevelopmental, and psychiatric disorders. Accumulating evidence from both clinical and preclinical studies provides a growing body of support for the relationship between estrogen and estrogen signaling, and psychiatric and neurological conditions. Mounting evidence has indicated that ERs and estrogen are involved in the pathophysiology of several psychiatric and neurological disorders such as schizophrenia, bipolar disorder, MDD, autism spectrum disorder, attention deficit hyperactivity disorder (ADHD), anxiety disorders, eating disorders, substance use disorder, AD, and PD (for detailed reviews please see [83,88,89]).

Reduced levels of both ERα and ERβ in the CA1 hippocampal synapses in female rats have been reported as they age [88,90,91,92]. In another study, Hu et al. demonstrated that in female patients with AD, the number and proportion of nERα in the CA1 and CA2 areas of the hippocampus were reduced compared to the matched healthy controls [93]. On the contrary, the upregulation of ERα and ERβ, especially nERα, in the nucleus basalis of Meynert, vertical limb of the diagonal band of Broca, infundibular nucleus of the hypothalamus, and medial mammillary nucleus (only nERα and not ERβ) has been shown to be linked to the pathophysiology of AD when compared to age- and sex-matched controls [94,95,96,97]. Though some conflicting findings on the role of ERα in the pathophysiology of AD exist, ERα has been shown to play a role in AD risk and progression [88]. Nevertheless, studies on ERβ were more consistent, and in a study on the brains of females with AD, the downregulation of neuronal mitochondrial ERβ in the frontal cortex has been observed [98].

Moreover, both men and women with schizophrenia have been found to express lower levels of hippocampal dentate gyrus ERα [99], while no evidence in support of the involvement of genomic variations in ERα and ERβ genes in the etiology of bipolar disorder was found in two studies exploring the possible relationship [100,101].

We have only limited studies comparing the expression of ERs in the brains of patients with psychiatric illnesses such as bipolar disorder, generalized anxiety disorder, and ADHD. In one study, it has been revealed that the serum concentration of GPER in euthymic patients with bipolar disorder is higher than that of control subjects, and this was proven to be unrelated to the medications [102]. However, a decrease in the serum levels of patients with ADHD compared to healthy controls was found [103].

In both drug-naïve patients with generalized anxiety disorder and those with MDD, enhanced serum levels of GPER have been observed, positively correlated with anxiety and depression severity [104,105]. Moreover, the overexpression of ERα in the dorsolateral prefrontal cortex and anterior hippocampus of depressed male patients in comparison to female patients with MDD has been reported [99]. Finally, in autism spectrum disorder, the downregulation of ERβ in the middle frontal gyrus was observed [106], providing another line of evidence for the association of the ERβ gene with autistic traits [107].

ER knock-out animals have significantly contributed to our understanding of the crucial functions of ERs. However, they often fail to fully recapitulate the endogenous regulation and signaling of ERs and suffer from exerting multitude of undesirable outcomes in various tissues, including the disruption of hypothalamus-pituitary-gonadal axis. PET imaging can provide insights into the expression levels of receptors, and can serve as an even more valuable tool when combined with other techniques for a better understanding of the roles played by ERs in the pathophysiology of brain disorders. Furthermore, PET imaging offers the advantage of longitudinally monitoring changes in ERs in the course of a brain disease, or throughout its progression.

## 5. PET Imaging of ERs

To longitudinally visualize the dynamic interplay between ERs and the functional changes in the brain, as well as associated alterations in behavior, cognition, and emotion in neuropsychiatric disorders, PET imaging can be a valuable tool. Despite the availability of several PET tracers to study ERs, their utilization has been predominantly limited to oncology. In brain disorder studies, the current methods are often restricted to autoradiography and ex vivo tissue counting, rather than in vivo PET imaging [108].

16α-[^18^F]fluoro-17β-estradiol ([^18^F]FES) is the most widely used PET tracer to study ERs. There are incongruities among a few available studies using this tracer as a marker to quantify ER occupancy in the brain. It has so far been shown that [^18^F]FES is primarily suitable for studying the ERs of the brain regions with a high expression of ERs, such as the pituitary and hypothalamus [108].

In a rat study, the pituitary, hypothalamus, bed nucleus of the stria terminalis, and amygdala exhibited the highest uptake of [^18^F]FES in a descending order [109]. Khayum et al. [109] demonstrated that the uptake of [^18^F]FES was influenced by the estrous cycle and fluctuations of ovarian sex hormones. Ovariectomy increased the uptake of the tracer, while exogenous estradiol administration reduced its uptake in the pituitary and hypothalamus, the two areas where [^18^F]FES tracer was mostly detected due to a higher density of ERs [109]. The study also concluded that semi-quantitative standard uptake value analysis is prone to be more sensitive to the endogenous estrogens in blood, and thus quantitative kinetic analysis is preferred [109].

A subsequent human study in healthy postmenopausal women revealed significantly higher accumulation of [^18^F]FES tracer in the pituitary compared to other brain regions, which was diminished after the administration of an ER antagonist, Elacestrant, indicating the specific binding of the tracer to ERs [110]. None of the other regions showed a decrease in [^18^F]FES occupancy, implying that due to its high lipophilicity, non-specific binding is likely. Unlike preclinical studies, no changes in the hypothalamus in this clinical study were observed, warranting further experiments and research to replicate and explain this finding.

Additionally, in oncology, [^18^F]FES has shown utility in detecting and diagnosing brain metastasis in patients with ER-positive breast cancer or double primary cancer, which may have otherwise gone undetected [111,112,113]. Another PET tracer developed to study ERs is 4-fluoro-11β-methoxy-16α-[^18^F]-fluoroestradiol (4FMFES). 4FMFES and [^18^F]FES were extensively examined in a cross-species study comparing the brain uptake of these tracers in humans, mice, and rats [114]. 4FMFES yielded better contrast and lower non-specific accumulation [114] presumably due to its higher resistance to being metabolized and lack of binding to sex-hormone-binding globulins (SHBG) [115], despite its lower uptake in the pituitary. Both tracers were corroborated to have higher selectivity toward ERα than ERβ [116,117,118].

## 6. The Need for a PET Tracer

As our understanding of the diverse and crucial roles of ERs in the brain, and in particular, their connection to neuropsychiatric disorders continues to expand, there is an increasing demand for high-quality target-specific PET tracers for various ER subtypes. The current PET tracers utilized to detect ERs suffer from high non-specific binding as well as lack of selectivity. The discovery of different subtypes of ERs and their diverse pharmacology has highlighted the need for specific tracers that can selectively bind to, visualize these receptor subtypes in the brain, and provide us with a better understanding of their functions. Such tracers, in conjunction with non-invasive PET imaging, would enable researchers and clinicians to better understand the involvement of specific subtypes of ERs in different brain regions and their potential implication for neuropsychiatric disorders.

As ERs are involved in assorted physiological functions, such as neuroplasticity, neurotransmitter release, etc., the development of new selective PET ligands for distinct ER subtypes can enhance our ability to study the distribution, density, and function of each ER subtype in the brain in vivo. This valuable insight into the possible mechanisms underlying pathophysiology of neuropsychiatric disorders can boost the development of targeted therapeutic interventions.

## 7. Criteria for a Good CNS PET Tracer

When developing CNS PET ligands, the BBB must always be considered. Most CNS PET radiotracers pass through the BBB via passive diffusion rather than active transport [119]; therefore, good CNS PET tracers must be soluble enough in the lipid layer of the BBB and small enough to pass through the endothelial cell membrane to cross the BBB, implying that having proper lipophilicity is key [120,121]. LogP and LogD_7.4_ are values that can predict the lipophilicity of a molecule, where favorable BBB-permeable molecules have LogP values ranging from 2 to 4 and LogD_7.4_ values between 1.5 and 3.5 [122,123,124]. Topological polar surface area (TPSA), which is defined as the sum of surface area of polar heteroatoms of a molecule, is also another predictive tool to determine molecular structures that are considered favorable for BBB permeability [123,124]. Fortunately, most estrogenic compounds are naturally lipophilic enough, making the task easier. However, it is noteworthy to mention that the BBB’s efflux transporters, specifically P-glycoprotein (P-gp) must also be taken into account. A molecule may have sufficient permeability to cross the BBB, but could still be a substrate of P-gp, thereby potentially resulting in the poor uptake of a CNS-penetrant PET tracer [123]. One strategy to overcome this issue in the case of having a promising radiotracer is to co-administer a P-pg substrate PET tracer with an inhibitor of this transporter, such as cyclosporin-A or elacridar [125]. In PET imaging, solubility is hardly a concern and is infrequently assessed [126].

Favorable pharmacokinetic parameters for a potential CNS PET tracer are also of paramount importance. The intravenous injection of PET tracers removes the absorption and excretion from ADME (absorption, distribution, metabolism, and excretion) consideration, and challenges only lie in metabolism and distribution [123]. CNS PET radiotracers need to exhibit favorable tissue distribution and time of activity. This ensures that a radiotracer is readily taken up and is rapidly cleared from non-target tissues to provide sharp imaging contrast. Most CNS PET tracers typically require a volume of distribution at steady state (Vss) greater than the total body water volume. However, it is critical to bear in mind that high Vss values can sometimes be linked to non-specific binding or tracer accumulation in muscle, skin, or fat tissues [123]. Thus, achieving a balance is crucial. On the other hand, slow kinetics can result in less informative images, and a propitious time of activity is essential for scheduling imaging acquisition procedures and the interpretation of results. Furthermore, micro-dosing of PET tracers alleviates concerns about the potential toxicity of the metabolites [126]. Nonetheless, CNS PET tracers should be designed with careful consideration to avoid having biologically active radio-metabolites in the brain [124]. Metabolites also need to be BBB-impermeable to prevent any interference with the integrity of the imaging signal [124]. A special caveat with PET tracers labeled with ^18^F is the accumulation of fluorine ions in the skull due to defluorination, resulting in the spill-over of radioactivity, compromising the accuracy of binding quantification [124]. (For more information in this regard please refer to [124]).

The success of developing a PET tracer, however, does not only boil down to the tracer itself. The biological molecular target is also of utmost importance for the successful development of a novel PET tracer. The expression level of the target receptor and a propitious profile of brain bio-distribution determine whether a PET ligand could be specific enough. PET tracers need to be highly (adequately) potent and selective, and occupy the same binding site as the drug or the endogenous ligands of interest. Potencies in sub-nanomolar and nanomolar concentration ranges are favorable [120,121]. High target expression and high affinity toward that target can be quantified by an index, B_max_/K_d_, which should ideally be equal or more than 10 [120,121] and can compensate for one another. If a target of interest has promising pharmacokinetic parameters, but a low expression level, then it needs to have higher affinity (long target occupancy) to compensate for that, and vice versa [126]. An analysis of 20 studies of multiple successful PET tracers targeting various receptors in different brain regions of healthy human subjects reveals that the B_max_ values typically fall within the range of 0.7 to 103 nM, while the K_d_ values exhibit variation within the range of 0.02 to 6.1 nM. Notably, all of these tracers consistently maintain a B_max_/K_d_ ratio exceeding 3 [127].

Plasma protein binding is also something essential to think of upfront [123]. Determining the bound fraction of a radiotracer, which can impact its ability to cross the BBB, might be advantageous, especially when a reference region in the brain is not utilized. Estrogen binds both albumin and SHBG in humans [109]. While binding a radiotracer to plasma proteins shields it from peripheral metabolism, it can also influence the uptake of a PET tracer into the brain. It is worth noting that rats lack SHBG, which should be considered when interpreting between species findings [109].

Last but not least, radioligands should possess the proper chemical structure for the incorporation of either [^11^C] or [^18^F], and this process should preferentially be the final step in the synthesis and purification of the radioligands considering the short half-lives of these tracers, which are 20 and 110 min, respectively [120,121]. For a summary of the factors mentioned please refer to (Box 1).

Box 1Criteria for a successful potential CNS PET radiotracer.Molecular weight (MW) of less than 500 Da2 < LogP < 41.5 < LogD_7.4_ < 3.5Less than 3 hydrogen bond donorsLess than 9 hetreroatomsTopological surface area of less than 90 Å^2^Favorable pKa of functional groupsSubnanomolar to nanomolar affinity for targetTarget selectivityLack of brain penetrant metabolitesFavorable kineticsNot a substrate for P-gp

## 8. Setback for Detection of ERs in the Brain

In order to explore the brain regions with lower density of ERs, PET tracers with higher affinity to ERs are needed. In addition, tracers that have a high tendency to remain bound to the SHBG and albumin are sequestered in blood circulation (bound fraction) and will not be able to reach their target in the brain.

B_max_ and bio-distribution are particularly important in the case of mERs, which structurally resemble nERs and account for less than 5% of the population of ERs. Additionally, given the generally low expression profile of ERs in most brain regions, the development of CNS PET tracers for ERs should heavily depend on achieving high molar activities. Another challenge lies in discriminating GPER signaling from ERα than ERβ signaling, as most ligands lack selectivity for ERα or ERβ.

In the following subsections, we shift our attention to various selective ligands that have been developed, and we propose the labeling of some of these ligands as PET tracers. To assess the suitability of the suggested compounds as potential CNS PET traces, we conducted computational simulations on ADME parameters using SwissADME (http://www.swissadme.ch/).

### 8.1. ERα or ERβ

Recently, a selective compound, AB-1, has been identified, demonstrating selectivity toward ERα and ERβ, but not GPER, enabling us to exclude any potential response initiated by GPER and restrict it to ERα and ERβ signaling [128] (Figure 2).

Furthermore, estetrol (E4) is another naturally occurring estrogen produced exclusively by the human fetal liver during pregnancy, which has been reported to activate nERs, lacking the capability of activating mERs (Figure 2) [129,130]. Abot et al. have also shown that in the presence of E2, E4 antagonizes membrane-initiated signaling [129]. Therefore, it can be exploited as a potential PET tracer to discern nERs from mERs.

Additionally, distinction between ERα and ERβ seems essential in delving into the distinct roles each receptor plays in modulating CNS function and the pathophysiology of neuropsychiatric disorders. By understanding the specific contributions of ERα and ERβ, we can gain deeper insight into their modes of action in various brain disorders. Several molecules were demonstrated to exhibit selectivity toward ERβ.

Darylpropionitrile (DPN), indazole chloride, WAY-166818, WAY-200070, LY500307, LY3201, and ERB-041 are all highly potent and selective molecules with a preference for ERβ over ERα, and have been tested in models of various brain disorders [131]. Among them, compounds DPN, WAY-200070, LY3201, and ERB-041 have the potential to either be directly labeled with either ^11^C or ^18^F (DPN, LY3201 and ERB-041), or by exchanging Br with ^18^F (WAY-20070), and serve as structurally modified compounds to develop PET radioligands (Figure 3). Recently, a radiolabeled analogue of ERB-041, [^18^F]-PVBO, has been developed, showing selectivity toward ERβ [132]. However, it suffers from moderate defluorination due to having a labeled fluorine atom bound to an aliphatic carbon, and a subsequent accumulation of the tracer in the bones has been reported in mice [132]. This could be especially problematic in the case of imaging ERs in the brain because of its accumulation in the skull.

Propyl pyrazol triol (PPT) is a compound synthesized and developed specifically to selectively activate ERα. It exhibits a potency that is more than 1000-fold higher towards ERα compared to ERβ [133,134]. The compound was designed to target and activate ERα specifically, allowing for the investigation of the receptor’s functions and effects [133]. On the other hand, methyl-piperidino-pyrazole (MPP) was developed by the same research group with the intention of achieving the antagonism of ERα [135]. Both PTT and MPP can serve as a backbone for the development of PET tracers specifically selective to ERα (Figure 4). The incorporation of suitable radionuclides into their structure may provide the non-invasive visualization and quantification of ERα dynamics in the brain and brain disorders. PTT and MPP may not be labeled directly either with ^11^C or ^18^F. Here, labelling might be achieved by coupling a chelator to the PTT or MPP backbone and using either ^18^F or ^68^Ga. However, ADME prediction suggests that none of these compounds can penetrate the BBB.

### 8.2. Nuclear Receptors outside the Nucleus: mERα and mERβ

As mentioned earlier, the structural similarity between mERs to nERs makes it arduous to develop specific ligands to detect ERs located at the membrane. Furthermore, the low abundance of mERs, accounting for virtually 5% of the total ER population [136], is another complication to the development of a selective tracer. However, there are several strategies to selectively target mERα and mERβ.

The resolution of PET imaging does not allow for distinguishing between intracellular and extracellular receptors. Nevertheless, various strategies may be employed to potentially be able to differentiate between receptors located intracellularly or at the plasma membrane. These strategies include the designing of radioligands that are unable to penetrate the plasma membrane, developing ligands with preferential selectivity toward membrane-bound receptors, and combining PET with other techniques, such as two-photon excitation microscopy to verify data obtained by PET imaging.

One approach involves structural modification of some steroidal and non-steroidal compounds with the aim of lowering their affinity for ERs, which leads to an enhanced selectivity toward non-genomic (extra-nuclear) signaling over genomic (nuclear) signaling [137]. These modified compounds, known as pathway preferential estrogens (PaPEs), represent selectivity toward mERs [137,138,139,140] while maintaining a suitable lipophilicity profile, making them hold promise as potential candidates for developing PET tracers. PaPE-1, PaPE-2, and PaPE-3 have been designed [137] to exhibit preferential selectivity to mERα and m ERβ rather than nERs by lowering the binding affinity of these compounds to both ERα and ERβ and increasing their dissociation rate from ERα [137]. Considering the abovementioned criteria from a chemical point of view for developing CNS PET tracers, PaPE compounds can potentially be labeled as a PET tracer for detection of mERs (Figure 5). Among these compounds, both have been predicted to be capable of crossing the BBB, but only PaPE-2 was predicted to be a non-substrate of P-gp.

Another strategy is to covalently conjugate potential candidates with dendrimers, such as (polyamidoamine) PAMAM, which are able to cross the BBB but are unable to penetrate the nucleus due to their large size and positive charge. Estrogen-dendrimer conjugates (EDCs) have been utilized to differentiate between membrane-initiated estrogen signaling from that of nucleus [141,142] (Figure 5). Surfaced-engineered dendrimers labeled with various radio-nuclei such as ^68^Ga, ^64^Cu, ^18^F, ^89^Zr, etc., have recently been developed, offering new possibilities for imaging a target with improved profile and with applications in brain disorders [143,144,145,146,147,148,149]. There have been reports indicating that labeled PAMAM dendrimers can cross the BBB upon intravenous and intra-arterial administrations [148,150]. However, it should be noted that the penetration of heavy metals into the BBB is generally challenging. Another caveat is that EDCs undergo a significant morphological transition in response to changes in pH, rendering the estrogen inaccessible and consequently masking its bioactivity [151].

### 8.3. GPER

Compounds G-1 and G-15 serve as the selective agonist and antagonist of GPER, respectively, and can be exploited as a backbone for synthesizing radioligands specific to GPER detection. Several first-generation non-steroidal ^99m^Tc-labeled selective GPER radiotracers (confirmed by in vivo competition studies) (Figure 6) were developed, demonstrating binding affinities of 10 to 30 nM [152].

These radioligands belong to a family of neutral M(I)-tricarbonyl complexes (M = Re, ^99m^Tc) where the pendant tetrahydro-3*H*-cyclopenta[c]quinolone scaffold is coupled through linkers of different length and nature to pyridin-2-yl hydrazine and picolylamine bifunctional chelators. It has been unveiled that complex linkage with the hydrogen bond acceptor ethanone group resulted in the agonistic activity of the compounds, while triazole-linked complexes functioned as antagonists [153]. Moreover, the analogues needed to be neutral and uncharged to interact with the functional intracellular receptor and initiate the rapid signaling associated with this transmembrane GPER [154,155]. The steric volume of the conjugates also played a role, with reduced affinities observed as the steric volume increased [153].

However, more structural optimizations are necessary to incorporate either ^11^C or ^18^F for improved brain bio-distribution and imaging characteristics. Albeit these compounds benefit from favorable lipophilicity to be able to cross the BBB, it is hard to predict whether they could be potential substrates for P-gp, and further experimental investigations are warranted. According to our simulation, nonetheless, both are likely to be substrates of P-gp.

## 9. ADME-Driven Informed Selection of Ligands

In PET neuroimaging studies, antagonists are predominantly used, as they are capable of binding to both coupled and non-coupled G-protein-coupled receptors (GPCRs) with equal affinity, allowing for the imaging of the overall density of a given receptor. However, when it comes to agonists, they distinguish the active state of a protein from its inactive one, since they exhibit different affinities toward different states, resulting in a lower signal-to-noise ratio [156]. Another challenge with PET radiotracers designed as agonists is that the rate at which a receptor converts from high-affinity (active) to low-affinity (inactive) status (which has been demonstrated to occur upon binding of an agonist) can give rise to rapid dissociation of the radioligand from the receptor [156]. On the other hand, the use of agonist PET tracers can offer advantages by allowing the selective detection of active receptors in the brain [156,157]. This approach is particularly beneficial for exploring coupled and non-coupled receptors in vivo [156,157], as changes in the state of receptors in the brain under pathophysiological conditions have been reported in vitro [158]. Most of the ligands we suggested for radiolabeling are therefore agonists.

Based on the potential of already existing ligands for various subtype of ERs for having a radionuclide incorporated, we employed computational simulations on ADME parameters to evaluate the suitability of the proposed compounds as potential CNS PET traces using SwissADME [159]. Among the suggested compounds, E4, MPP, and PPT were the only ones found to be impermeable to the BBB, while the rest exhibited favorable physicochemical properties, enabling them to cross the BBB (Figure 7 and Table 1). However, among those BBB-permeant tracers, only WAY-200070, ERB-041, DPN, and PaPE-2 were predicted not to be substrates of P-gp (Figure 7 and Table 1). However, experimental data are needed to corroborate whether these compounds are effluxed by P-gp. Even when experimentally determined, it is essential to keep in mind that rodents display higher efflux transporter activity compared to pigs, primates, and humans [160], and therefore, when encountering unsatisfying results, it is important to thoroughly examine this aspect.

Based on these predictions, selective ligands for the detection of ERα are still lacking, but some alternative strategies such as the use of dual/multiple tracers can be explored. Assuming specific tracers are available for particular receptor subtypes, they can be employed to subtract the signal associated with the more selective tracer from the less selective one, where the less selective tracer can give a good indication of non-specific binding.

Such simulations, however, do not provide more detailed information on the metabolic stability of the tracer, kinetic modeling, binding affinity, and plasma protein binding of the tracer, all of which need to be determined experimentally.

## 10. Concluding Remarks

Despite the extensive use of PET tracers for ERs in oncology, there remain several unmet needs and areas that require further research to develop efficient PET tracers for the family of ERs. The presence of varied ERs in the brain and the local synthesis of E2 within neurons and astrocytes highlight the importance of understanding estrogen signaling in various functions and dysfunctions of the brain. The traces of estrogen signaling and ERs throughout the course of many brain disorders such as neurodevelopmental, neurodegenerative, and psychiatric disorders have put even more emphasis on developing better and more selective PET radioligands to non-invasively and longitudinally study ERs in a diseased brain. Although the precise mechanisms underlying the effects of estrogen signaling in brain disorders are still being elucidated, the visualization of ERs via PET imaging facilitates a deeper understanding of the pivotal roles they play.

Nonetheless, one of the key challenges is the development of highly specific and selective PET tracers for different subtypes of ERs in the brain. While there have been advancements in identifying and developing more selective ligands, more research is needed to optimize such ligands as potential PET tracers and assess their functionality, enabling accurate imaging and quantification of the expression and bio-distribution of the receptors.

Another area that requires further attention is the improvement of tracer stability and pharmacokinetic properties to be able to target specific subtypes of ERs in the brain. PET tracers ought to have sufficient stability to withstand the metabolic processes in the body, while having favorable pharmacokinetic properties to ensure adequate uptake and retention in the brain. Such parameters need to be experimentally determined.

Additionally, the translation of preclinical findings to clinical applications in the development of PET radiotracers involves careful considerations. It is essential to consider potential challenges when translating preclinical findings, mainly due to inherent biological differences among various species [161,162]. Special attention should be given to ensure that the target of interest is expressed in a comparable abundance and is bereft of any significant structural alterations across species. Additionally, with respect to kinetics, the rate of metabolism varies among different species, which may alter the translatability of findings [161,163,164]. The choice of a time point for a post-mortem study is also of importance, since dynamic processes are being evaluated, and this is a further argument for the need for validated in vivo imaging biomarkers for the ER field.

The choice of compatible radionuclides should balance the need for a long enough half-life, allowing sufficient imaging time, while at the same time being short enough to minimize radiation exposure. The efficacy, safety, and clinical utility of such proposed PET tracers also need to be further assessed.

Addressing these challenges will lead to the advancement of developing efficient PET tracers for peering into both nERs and mERs and gaining a better insight into the estrogen signaling mode of action in the brain.

## Figures and Tables

**Figure 1 biomolecules-13-01405-f001:**
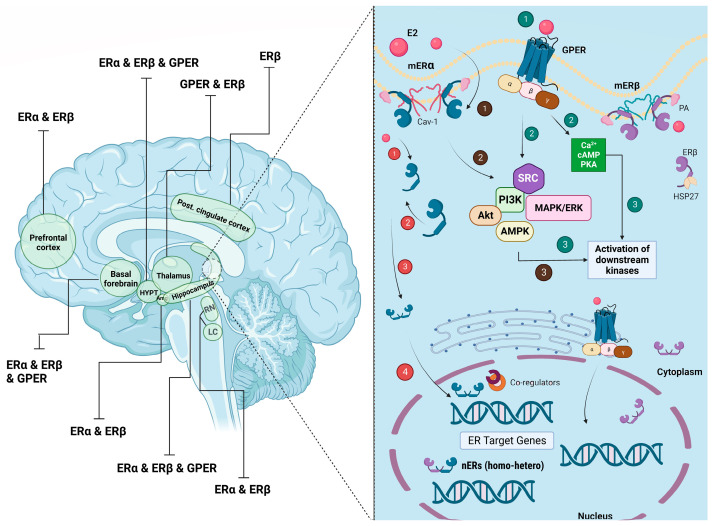
(**Left**) Localization of estrogen receptors in different brain areas. (**Right**) Membrane-initiated ER (mER) signaling vs. nuclear-initiated ER (nER) signaling. Once estrogen (E2) binds to its receptors (1), it initiates various cellular processes and responses depending on where and which ER is activated. mERα and mERβ are placed in the membrane raft (through palmitoylation of ERs) and scaffolded with caveoline-1 (Cav-1). Activation of various kinases and G proteins through physical interaction with mERα and mERβ (2), as well as GPER (2) leads to signal transduction and rapid physiological responses (within seconds) (3), whilst activation of intracellular ERs (1) causes the recruitment of monomeric ERs (2), which are bound to heat-shock protein 27 (HSP27) that subsequently form either homo- or hetero-dimers (3). Then, they are translocated to the nucleus to alter gene transcription (4). This process is much slower compared to cellular responses associated with mERs. Green and dark maroon circles represent the processes associated with the activation of membrane ERs, and light red circles depict events following the activation of nuclear ERs.

**Figure 2 biomolecules-13-01405-f002:**
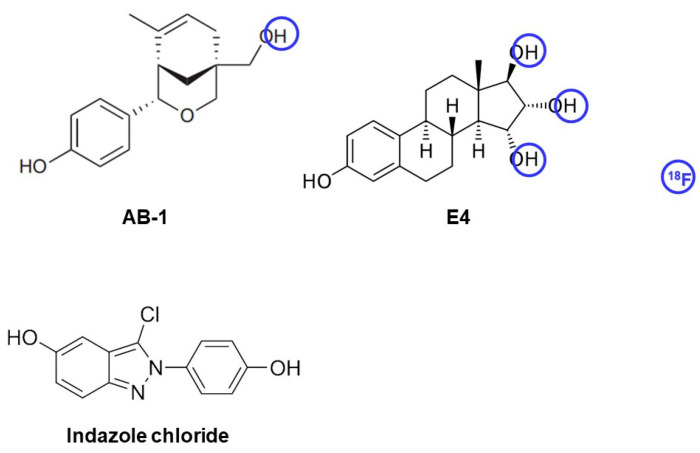
Potential ligands that into the structure of which a radionuclide can be incorporated, which is needed to develop PET tracers for either distinguishing ERα and ERβ from GPER (AB-1), or nERs from mERs (E4). Structural modification(s) are shown with blue circles. However, E4 lacks sufficient lipophilicity to cross the BBB.

**Figure 3 biomolecules-13-01405-f003:**
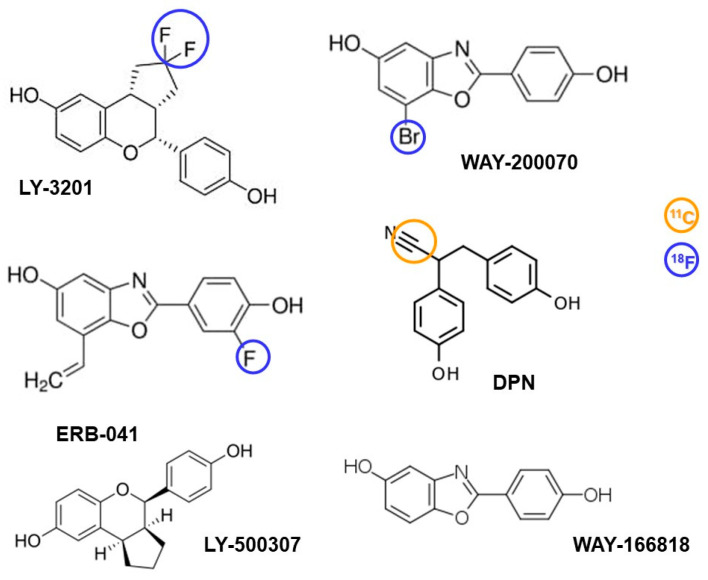
Potential ligands with possible structural modification(s) needed to develop PET tracers preferentially selective for ERβ. Suitable structural modifications are shown with blue and orange circles. All 4 structures with potential for being radiolabeled are predicted to be BBB-permeable, and LY-3201 is predicted to be the only substrate of P-gp of this grouped category.

**Figure 4 biomolecules-13-01405-f004:**
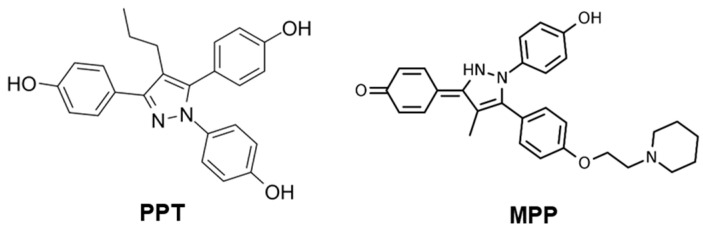
Ligands selective for ERα (PPT and MPP) are both predicted to be unable to cross the BBB. Moreover, incorporating a radionuclide in their structure is arduous.

**Figure 5 biomolecules-13-01405-f005:**
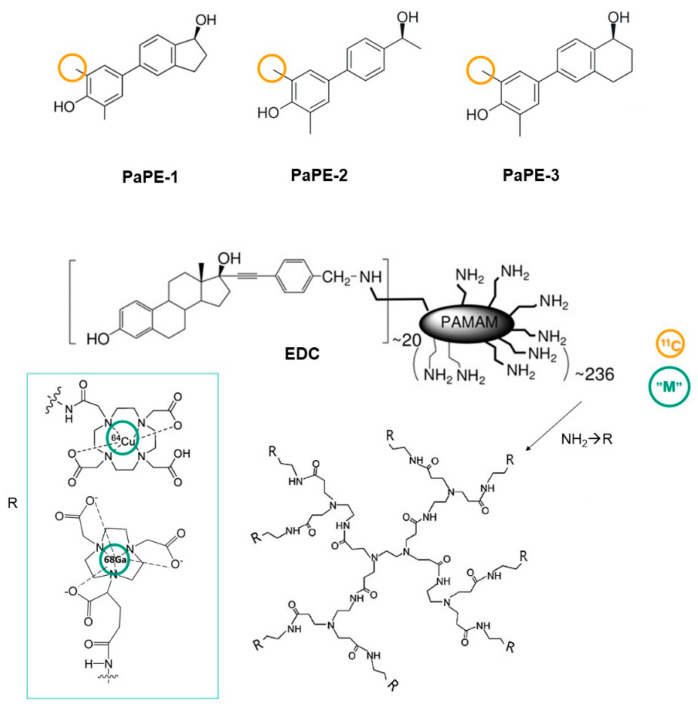
Potential ligands and the structural modification(s) needed to develop PET tracers preferentially selective for mERs. PaPEs can be labeled with ^11^C and have previously been shown to preferentially activate mERα. Estrogen-dendrimer conjugates (EDCs) [141] can be radiolabeled using complexes of radioactive Zr, Ga, or Cu. (Cu- and Zr-labeled PAMAM have been demonstrated to be BBB-permeable.)

**Figure 6 biomolecules-13-01405-f006:**
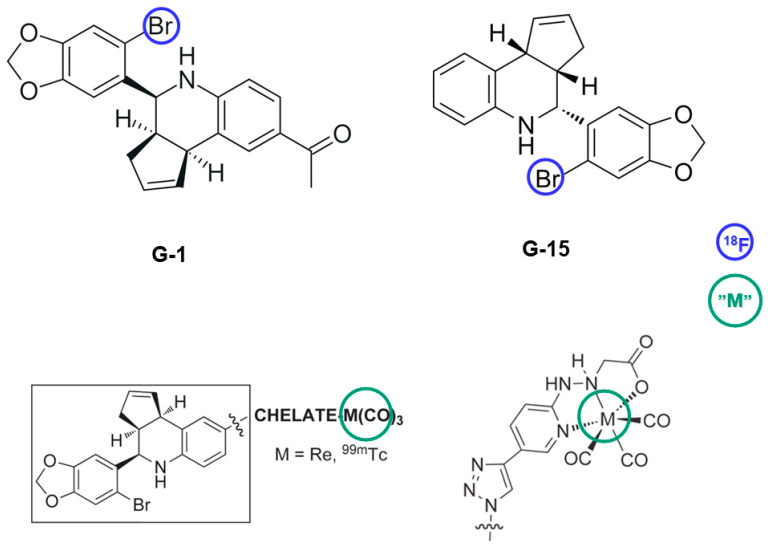
Potential ligands and the structural modification(s) needed to develop PET tracers selective for GPER. G-1 and G-15 are BBB-permeant, but might be subject to efflux by P-gp. The choice of metal to be able to traverse the BBB is also important.

**Figure 7 biomolecules-13-01405-f007:**
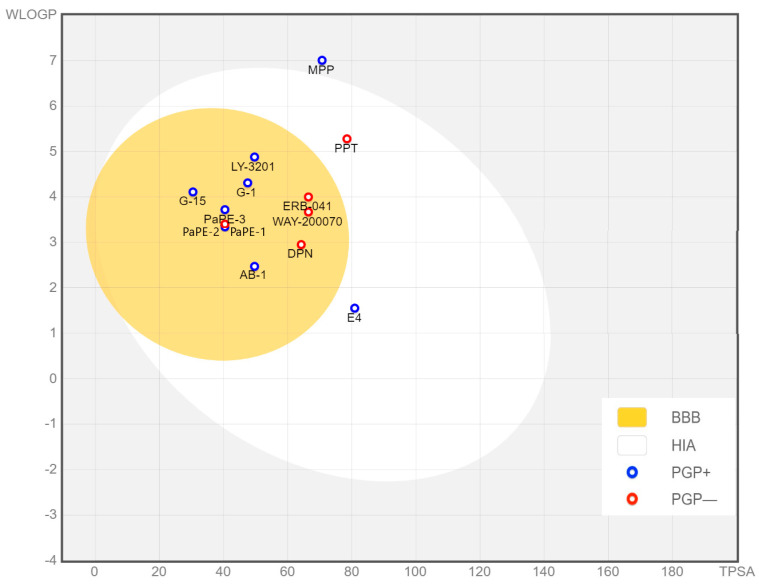
BOILED-Egg model. In this egg-shaped predictive classification plot, the yolk includes compounds with a high probability of BBB permeability. Blue dots show P-gp substrates (PGP+) and red dots for P-gp non-substrates (PGP−). White area represents passive gastrointestinal absorption (HIA).

**Table 1 biomolecules-13-01405-t001:** ADME parameters for the selected potential tracers predicted by SwissADME. Topological polar surface area (TPSA), LogP values, molecular weight (MW) and the number of H-bond donors and heteroatoms are shown. Based on such parameters, the permeability of the compounds of interest, as well as being potential substrates for P-glycoprotein, have been determined.

Molecule	Formula	MW	Heteroatoms	H-Bond Donors	TPSA	WLOGP	Consensus Log P	BBB Permeant	P-gp Substrate
AB-1	C_16_H_20_O_3_	260.33	3	2	49.69	2.47	2.27	Yes	Yes
E4	C_18_H_24_O_4_	304.38	4	4	80.92	1.55	1.66	No	Yes
LY-3201	C_18_H_16_F_2_O_3_	318.31	5	2	49.69	4.88	3.55	Yes	Yes
WAY-200070	C_13_H_8_BrNO_3_	306.11	5	2	66.49	3.67	2.93	Yes	No
ERB-041	C_15_H_10_FNO_3_	271.24	5	2	66.49	4	3.24	Yes	No
DPN	C_15_H_13_NO_2_	239.27	3	2	64.25	2.95	2.52	Yes	No
PaPE-1	C_17_H_18_O_2_	254.32	2	2	40.46	3.33	3.37	Yes	Yes
PaPE-2	C_16_H_18_O_2_	242.31	2	2	40.46	3.4	3.33	Yes	No
PaPE-3	C_18_H_20_O_2_	268.35	2	2	40.46	3.72	3.66	Yes	Yes
G-1	C_21_H_18_BrNO_3_	412.28	5	1	47.56	4.31	4.06	Yes	Yes
G-15	C_19_H_16_BrNO_2_	370.24	4	1	30.49	4.11	4.08	Yes	Yes
PPT	C_24_H_22_N_2_O_3_	386.44	5	3	78.51	5.28	4.25	No	No
MPP	C_29_H_33_Cl_2_N_3_O_3_	542.5	8	2	70.75	7.01	4.62	No	Yes

## Data Availability

Not applicable.

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
