# Peer review of "Peering into the Brain’s Estrogen Receptors: PET Tracers for Visualization of Nuclear and Extranuclear Estrogen Receptors in Brain Disorders"

_biomolecules, 2023, doi:10.3390/biom13091405_

Round 1

Reviewer 2 Report

“Peering into the brain’s estrogen receptors: PET tracers for visualization of nuclear and extranuclear estrogen receptors in 3 brain disorders”

The article is well written, although some of the references could be replaced by more current publications. However, I have concerns about the classification as perspective article. In my opinion a review article is the better choice, but the article would probably need to be extended in this case. For a view point, the article and the conclusion are too basic and the novelty or novel idea, view point, criticism etc. is missing. Although you give a very good overview about the existing tracer derivatives and their selectivity, I don’t necessarily draw the same conclusion of the need of more tracers without pointing out (subtarget-) specific scientific questions. However, beside the fact that these questions are not raised, I suppose also those could solely be answered with KO animal models. The actions of estrogens with regard to brain disorders are not limited to the expression of receptors, but act via genomic mechanisms, and therefore transcriptomic actions, which in fact cannot be imaged (pharmacodynamics). The authors have further mentioned as example  the expression of the driving enzyme for estrogen synthesis/conversion, aromatase, is sex and steroid-independent and don’t change during menstrual cycle. With PET, brain areas and target density or activity within a certain area (region or volume of interest (ROI or VOI) can be distinguished, which is definitely also of interest, however for a complex target as ER PET studies might be not the first choice as basis study design.

Potential improvements/general comments and questions:

For a perspective, also solutions of combination of the different ligands should be discussed as not all brain regions express the same subtypes as well as there are selective tracers available. Furthermore, can the authors imagine of solution using kinetic modeling or other methods or more sophisticated study designs (dual tracer concepts) ?

Line 27- 40: superficial and commonly known statement about CNS-PET, not related to ER. The relevance for the ER has to include e.g. kinetic modeling or the effect that PET can absolutely quantify if the studies are conducted dynamically including blood sampling, eventually including radio-metabolites analysis or generation of an image derived arterial input function

Line 53-55: the ability to cross the blood brain barrier (BBB) is general a challenge in CNS drug and PET tracer development. Currently, mostly the BBB score/CNS PET brain score are used for prediction (which is basically also a combination of your table). However, to set in relation to the presented ER ligands, BBB scores could have been mentioned or brain %ID, SUVs, K1 or different.  

Line 66: “…showing different physico-chemical and pharmacodynamic properties…” It sounds like you are referring to changes in binding etc. However, the plethora of pharmacodynamic action might be based on the function as transcription factor /genomic mechanism. I would recommend split this part and don’t mix structure descriptions of the derivatives with dynamics.

Line 69: Sentence redundant. Line 68 and 69 refer to local production or metabolism in the brain. Please reword/or delete.

Point 3: This part is important to explain the mechanistic pathways of ER, but not in relation to PET imaging and development of novel PET tracers. The resolution of PET will never allow any kind of distinguishing between extracellular and intracellular targets, also the limitation in half-life of the respective radionuclide for brain imaging is too short to see any long-term effects or target movement etc. Only a quantification in the different brain regions can be performed or distinguishing by the selectivity of the tracer, but only if the expression rate is of importance.

Line 114-116: “In 1975, the first evidence suggested the presence of nuclear estrogen receptors (ERα 114 and ERβ) at the cell membrane of endometrial cells [55,56], which, unlike the nuclear receptors, elicit an immediate response.”

Is it the cell membrane or did you mean here the nuclear receptor? Or is the membraneER internalized? Please explain or reword or correct typo.

Point 4:

Mentioned already a few times, but the mechanisms of ER in relation to several disorders seem to be based on the transcriptional potential and regulation of other gene expression through ER-Ligand complex (activation and inactivation of other pathways). Therefore, the actions of estrogens with regard to brain disorders might not be limited to the expression of receptors, but act via genomic mechanisms, which in fact cannot be imaged (pharmacodynamics). The authors have further mentioned that the expression of the driving enzyme for estrogen synthesis/conversion, aromatase, is sex and steroid-independent and changed for example in menstrual cycle were not visible using PET. With PET, only brain areas and target density or activity within a certain area (region or volume of interest (ROI or VOI) can be illustrated, that might not be enough for understanding estrogen metabolism and receptors.

Point 5: PET tracers

FES and FMFES already good tracers with a lot of studies? Why new developments? What is the concrete scientific question which can only be answered by PET? Again do you believe that the density or the transcriptional effects are the main driver of different diseases?

Point 7: Criteria for a CNS drug

That is a ground truth for all brain tracers, of course also ER PET brain tracers need to fulfill the common criteria of a “good” brain tracer. I think that is unnecessary info and is mentioned already in the introduction and here just a gap filler. Beside no BBB scores or the ability of the presented tracers to penetrate the BBB (or not) are mentioned. Please include that info.

Of the criteria for a “good” CNS PET tracer, only the LogP value is discussed in the text (289-291) as well as the ability for radiolabeling and the selectivity. Nothing is stated about the BBB penetration, or metabolic stability, binding affinities and unspecific binding of the presented ligands. Are there beside the lack of selectivity in some of the presented ligands, also other limitation?

Do you know if the ligands underly an active or passive membrane penetrating mechanism? (I suppose passive as the criteria for a good CNS PET tracer would partly be obsolete)

Line 297: we never speak of pharmacodynamics in PET, it is kinetics. Mostly Antagonist are designed, which do not show any dynamic effects. Furthermore, as mentioned by the authors, the sub-nanomolar concentrations are enough for the sensitive detection, but do not show pharmacological effects. Please delete “Pharmacodynamically speaking” or reword.

303- 310: Very importantly, you mentioned the Bmax/kd index, can you mention it for FES, FMFES or describe kd-values? However, in general most of the brain targets show low expression therefore most of the CNS drugs rely on high molar activities (and apparent molar activity). You might split this part from CNS criteria and add to point 8?!

316: Please explain what you mean by this sentence? Free fraction balance?

Point 8.1:

Inconsistent nomenclature of nER. Do you refer to nuclear ER with nER and ER and solely to mER as membrane. Are there other subtypes of GPERs or just the GPR30 known? (see also comment Line 114-116)

ERalpha and ERbeta can be distinguish in some brain regions, would that be enough for a plenty of studies? And in the thalamus ERalpha can be distinguished from beta and GPER. And so on, so what do you think about combination studies (multi-tracer concepts)?

Fig3: can you label the binding moiety or the structural elements most important for binding, as there are also other possibilities to introduce radiolabels on that molecules.

Paragraph 355-370/including fig 4:

PaPE-1, -2, and 3 are your designs? Please explain in this case as it is just mentioned that PPT and MPP can serve as basis for new developments, but nothing is explained further nor are those three ligands mentioned in the text or at a different paragraph (please think about re-structuring those parts or not divide them by sub-headings.

It is unclear for the reader, without checking the references, which of those tracers are existing, in pre- and clinical studies or designed by you.

Also, for these structures and in this paragraph and figure radiometals does not make sense to mention at all.

Point 8.2:

See comments regarding re-structure above.

Reference missing for PAMAM to penetrate the BBB, please add, especially because it is a Ga-68 ligand penetrating the brain. Anyways the radionuclide of choice in sense of energy and resolution would be F-18 or C-11, rather than Ga-68.

PAMPAM has nothing to do with the selectivity, but the implementation of other labeling strategies which does potentially not affect BBB penetration. Do the other ligands, all penetrate the BBB or not?

Point 8.3:

399-403: For brain targeting or peripheral usage?

416: in general, there are not successful prediction models for efflux transporter substrates as well as it is unlikely that the chelator compound will enter the brain. Please cite if there is indication that they cross the BBB.

General comments:

1)      Figures might profit from tabular form indicating the selectivity/specificity towards the sub-targets

2)      You might think of extend to a review article and split the challenges and ligands with regard to selectivity, labeling strategies and BBB penetration.

3)      The whole perspective seems to me like an introduction of a master or doctoral thesis and is in my opinion not a perspective. You mention a lot of tracers, which are more then enough to do research on this understudied subject in vivo. In case some specific evidence occurs with a specific question which cannot be targeted by those ligands, then there is the need for further development. I don’t see this at the moment for the very complex ERs.  

Therefore, either this “perspective” needs to specify scientific questions. We are heading a major problem in radiopharmaceutical sciences: the lack of new targets and consequently developments, instead pharmacist and chemist develop the fold-time the pretty much same tracer with slights better characteristics (or even not that). A perspective should give new opportunities, new targets, a view nobody has published yet, or transferred from another research field to the radiopharmaceutical community.

4)      Nothing is stated about other parameters of the tracers (metabolic stability, plasma protein binding, unspecific binding, synthesis parameters (like good yield, molar activities?) although many of them are mentioned as basis for a good CNS PET tracer.

5)      In a perspective the extended part should be the future direction.

Round 2

Reviewer 1 Report

Thank you for your consideration of my comments in the revision of the manuscript.

Reviewer 2 Report

Thank you for adressing all questions and remarks.